# Feasibility of flow-related enhancement brain perfusion MRI

**Julian Glandorf**[1,2]*, **Filip Klimeš**[1,2], **Andreas Voskrebenzev**[1,2], **Marcel Gutberlet**[1,2], **Agilo Luitger Kern**[1,2], **Norman Kornemann**[1,2], **Nima Mahmoudi**[3], **Mike P. Wattjes**[3], **Frank Wacker**[1,2], **Jens Vogel-Claussen**[1,2]

1 Institute for Diagnostic and Interventional Radiology, Hannover Medical School, Hannover, Lower Saxony, Germany, 2 Biomedical Research in Endstage and Obstructive Lung Disease Hannover (BREATH), Member of the German Centre for Lung Research (DZL), Hannover, Lower Saxony, Germany, 3 Institute for Diagnostic and Interventional Neuroradiology, Hannover Medical School, Hannover, Lower Saxony, Germany

* glandorf.julian@mh-hannover.de

## Abstract

### Purpose

Brain perfusion imaging is of enormous importance for various neurological diseases. Fast gradient-echo sequences offering flow-related enhancement (FREE) could present a basis to generate perfusion-weighted maps. In this study, we obtained perfusion-weighted maps without contrast media by a previously described postprocessing algorithm from the field of functional lung MRI. At first, the perfusion signal was analyzed in fast low-angle shot (FLASH) and balanced steady-state free precession (bSSFP) sequences. Secondly, perfusion maps were compared to pseudo-continuous arterial spin labeling (pCASL) MRI in a healthy cohort. Thirdly, the feasibility of the new technique was demonstrated in a small selected group of patients with metastases and acute stroke.

### Methods

One participant was examined with bSSFP and FLASH sequences at 1.5T and 3T, different flip angles and slice thicknesses. Twenty-five volunteers had bSSFP imaging and pCASL MRI. Three patients with cerebral metastases and one with acute ischemic stroke had bSSFP imaging and were compared to T1 post-contrast images and CT perfusion. Frequency analyses, SNR and perfusion contrast were compared at different flip angles and slice thicknesses. Regional correlations and Sorensen-Dice overlap were calculated in the healthy cohort. Dice overlap of the pathologies in the patient cohort were calculated.

### Results

The bSSFP sequence presented detectable perfusion signal within brain vessel and parenchyma together with superior SNR compared to FLASH. Perfusion contrast and its cortico-medullary differentiation increased with flip angle. Mean regional correlation was 0.36 and highly significant between FREE maps and pCASL and grey and white matter Dice match were 72% and 60% in the healthy cohort. Pathologies presented good overlap between FREE perfusion-weighted and T1 post-contrast images.

**Data Availability Statement:** All relevant data are within the paper and its Supporting Information files.

**Funding:** The authors received no specific funding for this work.

**Competing interests:** The authors have declared that no competing interests exist.

## Conclusion

The feasibility of FREE brain perfusion imaging has been shown in a healthy cohort and selected patient cases with brain metastases and acute stroke. The study demonstrates a new approach for non-contrast brain perfusion imaging.

## Introduction

The global burden of neurological diseases leads to enormous interest in sensitive and fast imaging methods for early detection and surveillance of numerous life-threatening conditions [1,2]. Especially brain perfusion imaging plays a key role in the diagnosis and patient management of patients presenting with ischemic strokes and brain tumors allowing to estimate tissue at risk prior to thrombectomy or surgery [3–6]. Various established techniques for regional brain perfusion mapping are used in clinical practice based on CT, MRI, SPECT or PET. However, the majority of applications are based on intravenous contrast media or radiotracers [7–9].

In the current clinical routine, dynamic susceptibility contrast (DSC) and dynamic contrast enhanced (DCE) MRI are established methods to image brain perfusion. They are based on the use of intravenous contrast media and tracer kinetic models [10]. Currently, the only available contrast- and radiation-free technique to image regional brain perfusion is arterial spin labeling MRI, which exploits magnetically labeled inflowing spins as an endogenous tracer to reduce the signal of the imaging plane according to its perfusion [11]. Unfortunately, low signal-to-noise ratio (SNR), susceptibility-related distortion, flow and motion may impair image quality and result in relatively long acquisition times.

In image acquisition of spoiled gradient echo Fast Low Angle Shot (FLASH) or balanced steady-state free precession (bSSFP) sequences, multiple RF pulses are applied within short time intervals. As a result, the longitudinal magnetization is not able to recover completely between the upcoming pulse repetitions and the signal decreases to a steady-state level. At this constant signal level, the regain of longitudinal magnetization equals the amount of transverse magnetization evoked by the next RF pulse [12,13]. During each heartbeat, saturated spins are replaced by unsaturated spins within the imaging plane, which leads to flow-related enhancement. The enhancement occurs at the rate of the heart frequency and–once detected–the images can be sorted according to their phase within the heart cycle and the phase of the pulse wave can be calculated [14]. Consequently, a voxel with high perfusion and consequently higher inflow of fresh spins exhibits a higher signal amplitude variation compared to a voxel with low perfusion.

Similarly to time-of-flight (TOF)-MR-Angiography, flow-related enhancement is the basis of perfusion measurements in MRI techniques related to Fourier decomposition (FD) like phase-resolved functional lung (PREFUL) MRI, which provides various ventilation- and perfusion-weighted parameters of the lung during free breathing [14–17]. These techniques have demonstrated their value to detect and monitor different lung diseases [18,19] and it was shown to be feasible with the widely available FLASH sequence and with bSSFP sequences [20,21]. The applied PREFUL technique detects the pulse wave and is intrinsically very sensitive to both phase shifts and amplitude reductions by vascular occlusion or destruction, e. g. as in chronic thromboembolic pulmonary hypertension or potentially in ischemic strokes [14,22,23]. However, applications focusing on the central nervous system, particularly the brain, have not been examined so far.

This study consists of three experiments to investigate the feasibility of brain perfusion imaging by flow-related enhancement (FREE) MRI:

1st aim: Determining a feasible imaging protocol with optimized flip angle and slice thickness.

2nd aim: Evaluation of FREE MRI in a healthy cohort in comparison to pseudo-continuous arterial spin labeling (pCASL) MRI.

3rd aim: Evaluation of FREE MRI in patients with brain metastases and acute stroke in comparison to i.v. contrast-enhanced brain MRI or CT perfusion as clinical reference standards.

## Materials and methods

### Participants and imaging

Firstly, to determine a feasible imaging protocol, one healthy participant (male, 30 years old) underwent MRI using FLASH and bSSFP sequences at 1.5T and 3T, at different slice thicknesses and field strengths (Magnetom Avanto and Vida, Siemens Healthineers, Erlangen, Germany).

Secondly, to compare FREE and pCASL MRI, twenty-five healthy volunteers (11 female, 14 male, age range 19–61 years) had bSSFP imaging together with 2D pCASL MRI at 3T (Magnetom Vida).

Thirdly, to demonstrate the new perfusion mapping in a clinical setting, three patients with cerebral metastases (64-, 69- and 80-year-old males)–two of which with malignant melanomas and one patient with a pulmonary adenocarcinoma–and one patient with an acute ischemic stroke (male, 78 year old, last-seen-well 3 days previously, National Institutes of Health Stroke Scale 16 Points) were examined with bSSFP imaging. The perfusion maps were compared to T1 post-contrast MRI (Magnetom Verio and Aera, Siemens Healthineers, Erlangen, Germany) and with CT perfusion imaging (Aquilion One, Canon Medical Systems Corporation, Japan), respectively [24].

S1–S4 Tables present the MRI and CT imaging protocol parameters. pCASL parameters were chosen according to Alsop et al. with matching resolution to the bSSFP images [25]. The study was approved by the institutional ethics committee and written informed consent was obtained from the participants.

All participants were scanned in head first supine position and 500 images of a single 2D slice were acquired. All images were interpolated to the final image matrix of 256 x 256 pixels prior to reconstruction via zero filling interpolation.

### Perfusion-weighted maps

MATLAB R2020b (The MathWorks, Natick, MA, USA) was used for postprocessing and analysis. The bSSFP images were evaluated on the basis of phase-resolved functional lung (PREFUL) MRI to generate the perfusion-weighted FREE maps [14]. Compared to the previous publication, a high pass filter at 0.5 Hz was applied to the images in order to exclude signal changes of lower frequencies. The cardiac phase of each time point was estimated by piecewise fitting the averaged signal of a brain vessel region of interest (ROI) to a sine function. The ROIs were manually drawn at the anterior cerebral arteries at the imaged slices. Phase information was calculated after sorting and interpolating the images. Perfusion-weighted (PW) maps presenting the maximum median amplitude of the full cardiac-cycle time series within the parenchyma ROI were created. Furthermore, temporal perfusion-weighted maximum intensity projections (MIPs) showing the maximal amplitude of the entire heart cycle for each voxel were calculated. Perfusion-weighted pCASL maps were generated based on subtractions of label and control images by the Siemens vendor toolkit using a 2D pCASL [26].

## Image registration

To achieve voxel-wise comparability, the FREE maps of the patients with metastases were registered towards the 3D T1 post-contrast images and the FREE map of the stroke patient was registered towards the mean transit time map of the perfusion CT. For this, average intensity projections of the post-contrast MR images were created to match slice thickness and slice position of the FREE images. The rigid registration was performed using advanced normalization tools (ANTs) [27]. To achieve full saturation and to ensure a steady-state, the first 20 images of every FREE data set were discarded.

## Image analysis

Frequency analyses were performed to detect perfusion signal within manually drawn vessel ROIs and parenchyma ROIs. Considering the non-central chi distribution in magnitude images of multi-channel receive coils, the temporal SNR was estimated from the ratio of the mean signal in a manually segmented ROI of the brain parenchyma and its standard deviation [28,29]. The perfusion contrast was calculated by the ratio of the perfusion signal amplitude and the mean signal of each voxel during the whole time series. The SNR and the perfusion contrast were compared for both sequences with different slice thicknesses and flip angles at grey matter and white matter ROIs.

## Correlation and spatial overlap between FREE and pCASL maps

Regional relationship between FREE maps and pCASL maps was evaluated by Pearson correlation analysis (S5 Fig). Binary FREE and pCASL maps were calculated using the 40th percentile of the perfusion values as a threshold according to the average volume fraction of gray and white matter in healthy individuals [30]. By this, voxels containing values above the threshold were considered as grey matter and voxels with lower values were considered as white matter. The spatial overlap was then assessed by Sorensen-Dice coefficients calculating the grey and white matter match and mismatch between FREE and pCASL maps. Sorensen-Dice coefficients are used to calculate the similarity of two samples and were calculated by summing the FREE- and pCASL voxels within the same compartment (grey or white matter), multiplying the sum by two and divide by the number of all FREE- and pCASL voxels combined. Sorensen-Dice coefficients were also calculated to demonstrate overlap of manually segmented metastases in FREE maps and T1 post-contrast images and the stroke area in FREE map and the mean transit time image of the CT perfusion, respectively. P-values $\leq 0.05$ were considered as significant. The data was presented as median values with the first and third quartile.

# Results

## Sequence evaluation

The frequency analyses of the bSSFP sequence presented a detectable perfusion associated peak at the frequency of the heart close to 1 Hz within the vessel and parenchyma ROIs (Fig 1). Although a strong perfusion peak was detectable in the frequency analysis of a vessel ROI in the FLASH sequence, it presented no perfusion peak in the parenchymal ROI (S1 Fig).

The bSSFP sequence provided about 3–9 times higher SNR than the FLASH sequence at 1.5T and up to 2.54 times higher SNR at 3T (Table 1). The difference of the SNR between grey matter and white matter in the FLASH sequence ranged between +11% to +17% at 1.5T and between +20% to +42% at 3T. The difference of the SNR between grey matter and white matter in the bSSFP sequence ranged between +21% to +23% at 1.5T and between +50 to +86% at 3T.

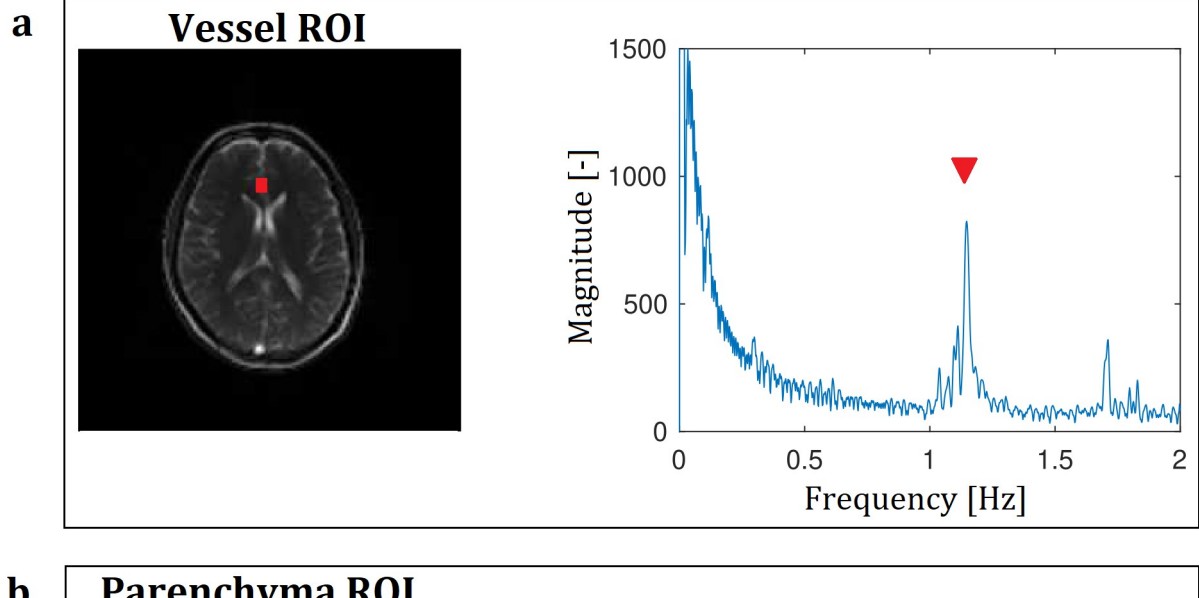

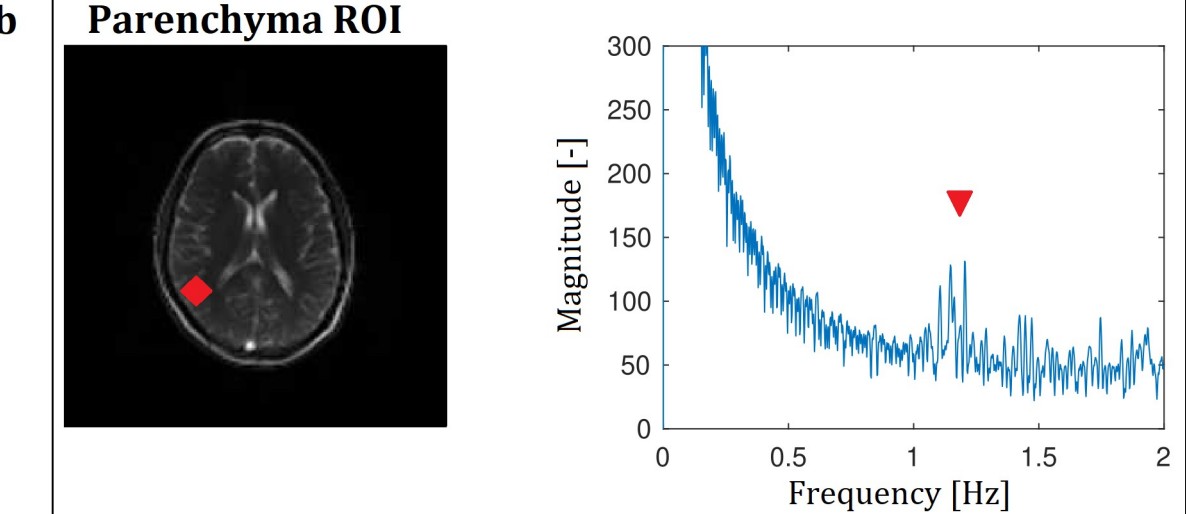

**Fig 1. Frequency analysis of a bSSFP.** A 50˚ flip angle indicating a perfusion related peak in the vessel (a) and in the parenchyma (b) ROI at the heart frequency close to 1 Hz (red triangles).

Although the FLASH sequence presented stronger perfusion contrast within vessels (S2 and S3 Figs), the bSSFP sequence presented a difference of the perfusion contrast between grey and white matter of up to +110% at 1.5T and of up to +463% at 3T (Table 1). For both sequences, an increase of the slice thickness led to an increase of the SNR, while an increase of the flip angle led to an increase of the perfusion contrast (Table 1 and Fig 2).

## Correlation and spatial overlap of FREE and pCASL - Healthy volunteer cohort

The healthy volunteer cohort provided a median voxel wise Pearson correlation of 0.36 between perfusion-weighted pCASL and FREE maps and between perfusion-weighted pCASL and FREE MIP. The correlations were highly significant with $p < 0.01$ (Table 1). FREE imaging took 230 seconds for one slice in each measurement.

**Table 1. SNR and perfusion contrast of the bSSFP (a) and the FLASH (b) sequences.** Varying slice thickness and flip angle in grey matter and white matter ROIs in a 30-year-old healthy volunteer at 1.5T and 3T. **Regional correlation and spatial Dice overlap towards pCASL-MRI in 25 healthy individuals (c).**

a)

| bSSFP | | 1.5T | | | | 3T | | | |
|---|---|---|---|---|---|---|---|---|---|
| Flip angle [°] | Slice thickness [mm] | SNR Grey Matter | SNR White Matter | Perfusion contrast x 10³ Grey Matter | Perfusion contrast x 10³ White Matter | SNR Grey Matter | SNR White Matter | Perfusion contrast x 10³ Grey Matter | Perfusion contrast x 10³ White Matter |
| 35 | 5 | 41 | 33 | 3.97 | 4.24 | 32 | 20 | 4.11 | 3.93 |
| 35 | 10 | 99 | 81 | 1.91 | 2.41 | 59 | 38 | 4.59 | 2.58 |
| 35 | 15 | 157 | 130 | 2.19 | 1.91 | 84 | 56 | 5.11 | 1.97 |
| 50 | 15 | 168 | 138 | 3.27 | 1.74 | 82 | 49 | 8.11 | 3.79 |
| 70 | 15 | 160 | 131 | 3.70 | 1.76 | 84 | 45 | 10.25 | 1.82 |

b)

| FLASH | | 1.5T | | | | 3T | | | |
|---|---|---|---|---|---|---|---|---|---|
| Flip angle [°] | Slice thickness [mm] | SNR Grey Matter | SNR White Matter | Perfusion contrast x 10³ Grey Matter | Perfusion contrast x 10³ White Matter | SNR Grey Matter | SNR White Matter | Perfusion contrast x 10³ Grey Matter | Perfusion contrast x 10³ White Matter |
| 5 | 5 | 12 | 11 | 15 | 17 | 25 | 20 | 2.81 | 4.01 |
| 5 | 10 | 26 | 23 | 7 | 8 | 49 | 41 | 1.77 | 2.94 |
| 5 | 15 | 39 | 35 | 6 | 8 | 73 | 61 | 2.21 | 2.44 |
| 20 | 15 | 28 | 24 | 12 | 18 | 32 | 23 | 15.27 | 38.00 |
| 40 | 15 | 18 | 16 | 15 | 30 | 49 | 37 | 6.36 | 10.62 |

c)

| Subjects | Voxel wise Pearson correlation | | Dice overlap | |
|---|---|---|---|---|
| 19 healthy individuals | pCASL* FREE$_{PW}$* | pCASL* FREE$_{MIP}$* | Gray matter match [%] | White matter match [%] |
| Median (1st and 3rd quartile) | 0.36 (0.21;0.44) | 0.36 (0.24;0.41) | 73 (66;76) | 60 (50;64) |
| | $p < 0.001$ | | | |

* Perfusion weighted (pw) pCASL-map; Perfusion weighted (pw) and maximum intensity projection (MIP) FREE map.

The median Sorensen-Dice overlap of the grey and white matter between the binary pCASL and FREE maps were 73% and 60% (Fig 3).

## Individual patient cases

A 64-year-old male patient with a metastasis of a malignant melanoma in his left cerebellopontine angle is shown in Fig 4A. The FREE perfusion-weighted map presents a Dice match towards the T1-post contrast image of 88% for the tumor area and of 92% for the brain area. FREE imaging took 127 seconds.

A 69-year-old male patient with a metastasis of a malignant melanoma in his right frontal lobe is shown in Fig 4B. The FREE perfusion-weighted map presents a Dice match towards the T1-post contrast image of 60% for the tumor area and of 99% for the brain area. A signal decrease within the right frontal lobe is detected in the FREE perfusion-weighted map after stereotactic radiotherapy with 24 Gray with a perfusion contrast of $1.92 \times 10^{-3}$ within the right frontal lobe and of $2.67 \times 10^{-3}$ within the left frontal lobe. Symmetrically low perfusion signal occurs in the posterior lobe within the vascular territory of the posterior cerebral arteries. FREE imaging took 127 seconds.

An 80-year-old male patient with a metastasis of a pulmonary adenocarcinoma in his right frontal lobe is shown in Fig 4C. The FREE perfusion-weighted map presents a Dice match

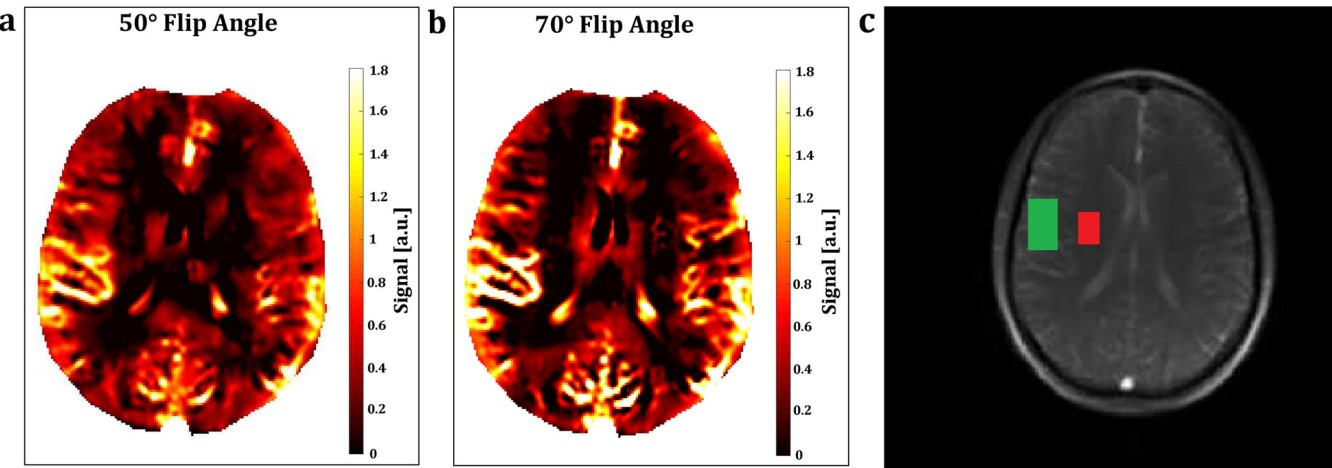

**Fig 2. Flip angle comparison.** Perfusion-weighted FREE maps of a healthy 30-year-old male participant with 50˚ (a) and 70˚ (b) flip angle, 15 mm slice thickness, 128 x 128 matrix size interpolated to 256 x 256, 250 x 250 mm FOV, 321 ms acquisition time for 1 image, 1 ms TE at 1.5T. The grey matter and white matter ROIs in the morphology map (c) (green and red boxes) showed a SNR of 167.80 and 138.02 and a perfusion contrast of 3.27 and 1.74 at 50˚ flip angle and a SNR of 160.27 and 130.83 and a perfusion contrast of 3.70 and 1.76 at 70˚ flip angle.

towards the T1-post contrast image of 52% for the tumor area and of 96% for the brain area. Notice the flow effect of the liquor in the FREE images in Fig 4A and 4C, whereas no CSF signal is visible in Fig 4B. FREE imaging took 127 seconds.

A 78-year-old male patient with an acute stroke of his left hemisphere is shown in Fig 5. The patient was last-seen-well 3 days previously and had 16 points on the National Institutes of Health Stroke Scale. Notice only minimal signal amplitude decrease in the temporal MIP, but clear demarcation of the stroke area in the parenchyma phase of the healthy hemisphere in the S4 Fig and the corresponding phase delay within the vascular territory of the occluded left middle cerebral artery in the S1 Movie. FREE imaging took 127 seconds for one slice.

## Discussion

In this study, we demonstrated the feasibility of phase-resolved FREE brain perfusion measurements. We determined the values of a bSSFP sequence to rapidly acquire images for further postprocessing allowing corticomedullary differentiation. The technique was evaluated in a healthy cohort and the feasibility was demonstrated in individual clinical cases.

Brain imaging has several favorable aspects for the application of FD-based techniques. Firstly, no time-consuming and possibly erroneous affine image registration process is necessary, because the brain presents only minimal movement and deformation. This avoids long postprocessing time and facilitates future implementations in urgent clinical scenarios like acute stroke as shown in our patient case. Secondly, brain tissue contains a higher spin density and is much more homogeneous than lung tissue causing less susceptibility artifacts [31–33]. This results in higher SNR, which can be traded for image acquisition speed and increased spatial and temporal resolution [34]. However, the assessment of potential advantages of higher field strength for brain perfusion measurements were beyond the scope of this study [35]. Of note, without repetitive shimming, the bSSFP sequence is vulnerable to severe banding artifacts, especially at 3T and at long TR [36].

Our results gave further hints to a feasible imaging protocol in regards to SNR and perfusion contrast. Firstly, although the FLASH sequence delivered strong perfusion contrast within larger vessels, its further use was withdrawn as no perfusion signal peak could be detected in the frequency analysis within the brain parenchyma. The depicted amplitude at the heart

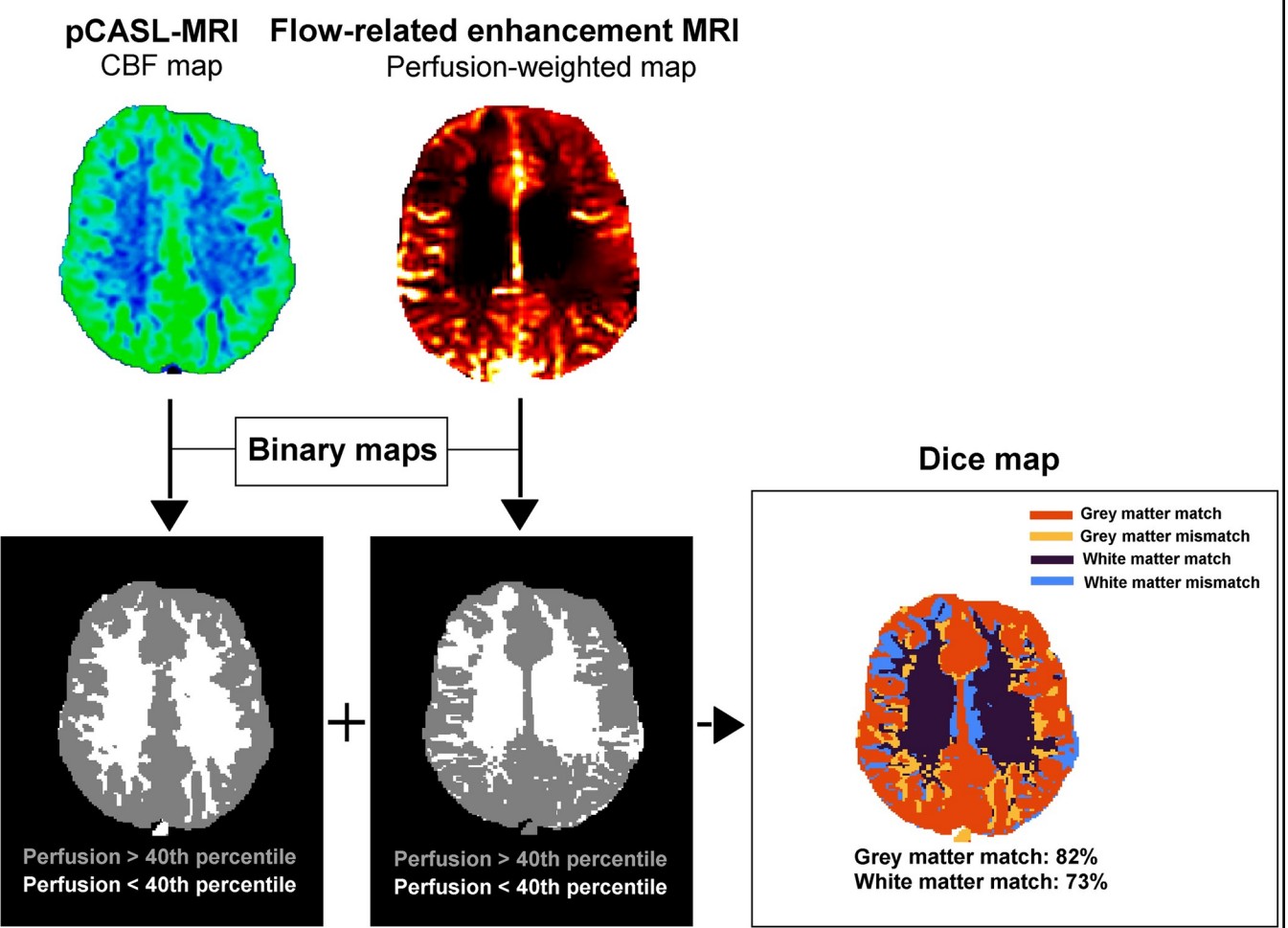

**Fig 3. Dice overlap of pCASL and FREE MRI.** Images of a 31-year-old healthy male proband. pCASL and FREE images are transformed into binary maps using the 40th percentile of its perfusion values as a threshold before calculating the grey and white matter overlap.

frequency is equal to the amplitude of noise (S1 Fig). Also, the FLASH sequence presented no obvious corticomedullary differentiation so that its parenchymal perfusion values may represent mostly noise. The values of perfusion contrast within the FLASH sequence are the result of a high amplitude of noise divided by a relatively low mean parenchymal signal causing this artifact. The flow-related signal enhancement in the bSSFP sequence increases with flip angle, because it affects spins depending on their velocity profile and their dephasing/repetition time [37]. This results in more signal increase of the spins within the systole increasing the amplitude of the flow-related enhancement. The presented differences of the perfusion contrast between the grey and white matter are within the range of previous reports using pCASL, DSC, DCE and $^{15}$O steady-state inhalation positron emission tomography [38–41].

Therefore, various circumstances have to be taken in account for optimal imaging parameters. Firstly, depending on the heart frequency, fast imaging speed needs to be achieved according to the Nyquist sampling theorem. This limits especially the resolution and flip angle together with the SAR limits. However, our results indicated, that a flip angle > 35˚ is sufficient to generate a corticomedullary differentiation within the physiological range as cited above. Secondly, a minimal slice thickness pronounces the perfusion contrast, but also increases noise. The maximum SNR of the bSSFP sequences was reached around 50˚ at 1.5T

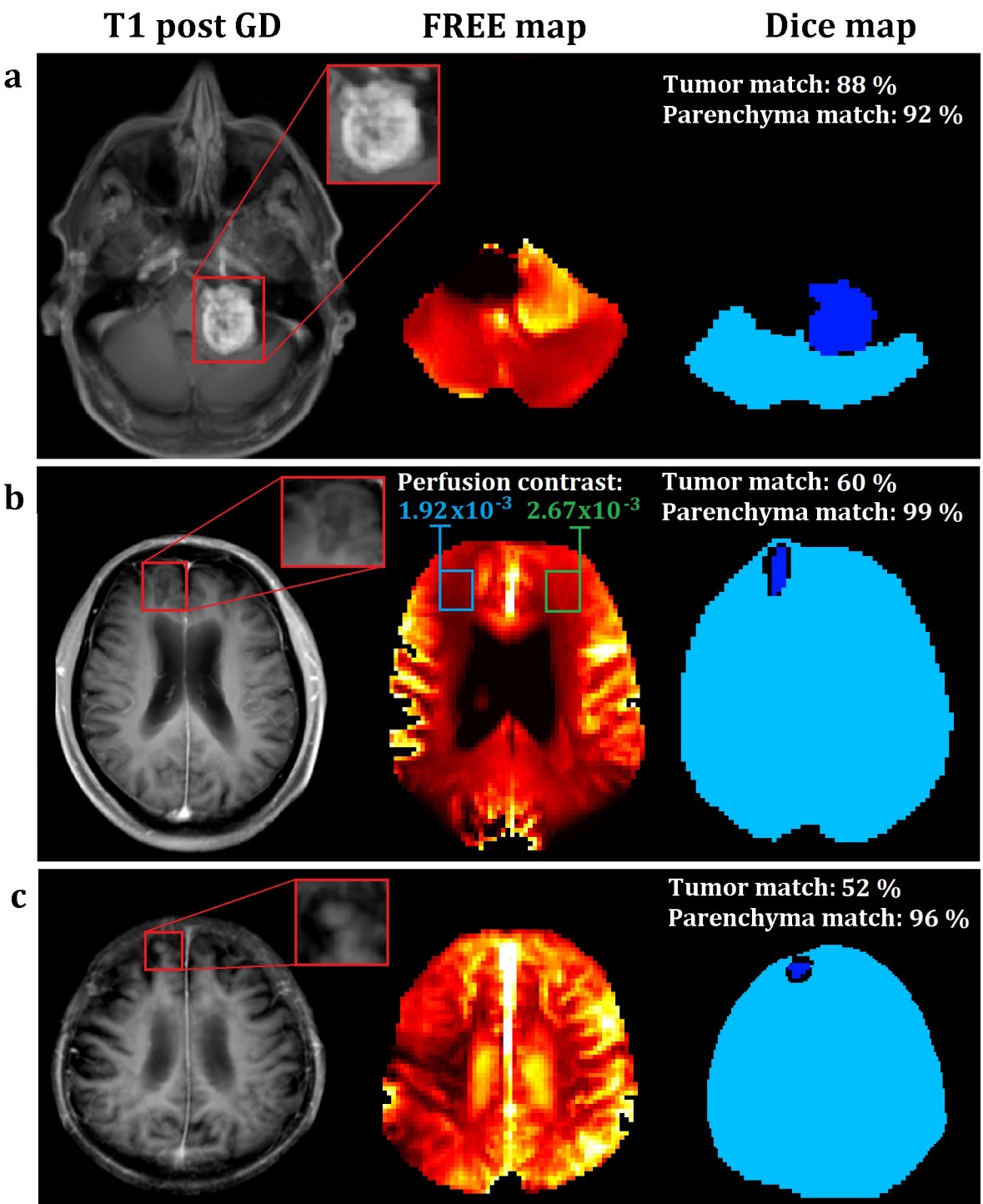

**Fig 4. Patients with metastases.** T1 post-contrast and perfusion-weighted images with the resulting Dice maps of three male patients with brain metastases of a malignant melanoma (a and b) and a pulmonary adenocarcinoma (c). Notice the differing perfusion contrast after radiation of the right frontal lobe in (b).

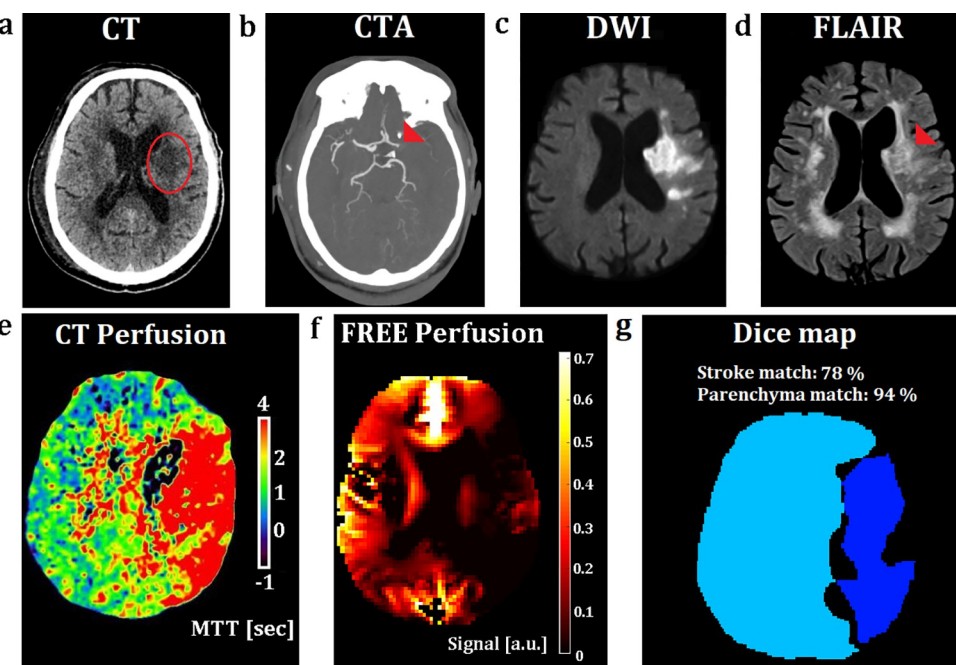

**Fig 5. Patient with an acute stroke.** CT and MRI images of a 78-year-old male patient with an acute M1 infarct of his left hemisphere. MRI was performed due to the unknown time window, while another emergency patient was treated. Afterwards this patient had thrombectomy, but showed a demarcated infarction as already visible on the previous CT and MRI. The demarcation is already visible on the native CT (a) and MRI fluid attenuated inversion recovery (FLAIR) (d) images (see red marking). CTA shows the occluded left middle cerebral artery (b) (see red triangle). Diffusion restriction is detected at the diffusion-weighted imaging (DWI) sequence (c). The CT perfusion (e) and the FREE perfusion-weighted (f) images present large Dice matching areas of 78% for the stroke area and 94% for the healthy brain area with delayed perfusion within the territory of the left middle cerebral artery (g).

and around 70˚ at 3T. All of these coherences need to be addressed depending on the clinical indication requiring imaging speed or perfusion contrast or resolution. Future developments in the new field of FD-based brain perfusion imaging should focus on the optimization of the imaging parameters for certain clinical indications.

The depicted inhomogeneities within the perfusion-weighted maps are also in part likely the result of the known dependency of FD-perfusion signal on the angle of blood flow towards the imaging slice and its degree of turbulence [42]. This causes pronounced perfusion values in the middle and anterior cerebral artery territories showing more orthogonal angles towards the imaging plane [43]. The limited number of visible arteries in some brain regions can also hamper the vessel ROI based sorting approach. Alternatively, a ROI could be placed at the superior sagittal sinus. A damped and complex waveform could potentially impair the phase sorting [44]. However, this has not been evaluated systematically yet.

Imperfect phase sorting together with low spatial resolution could reduce the differentiation between tissue perfusion and flow effects within the cerebrospinal fluid (CSF) in the subarachnoid space. On the other hand, visible flow effects suggest another potential application of this technique in the field of CSF circulation disorders [45,46].

Beside the amplitude of the flow-related enhancement, more sensitive and specific functional parameters need to be developed. Our results indicated the urgent need to determine phase-resolved perfusion including the phase shift of the pulse wave in clinical scenarios with vascular occlusion like ischemic strokes. Also, different ROIs for phase sorting need to be considered and evaluated by future studies pursuing quantitative perfusion measurements.

**Table 2. MRI brain perfusion techniques in comparison.**

|  | FREE MRI | ASL MRI | DCE MRI | DSC MRI |
|---|---|---|---|---|
| Whole brain analysis | Not yet | Yes | Yes | Yes |
| Perfusion quantification | Not yet | Yes | Yes | Yes |
| Dynamic information | Yes | No | Yes | Yes |
| Application of gadolinium contrast | No | No | Yes | Yes |
| Interstitial parameters | No | No | Yes | No |
| Degree of validation | Low | High | High | High |

Alternatively, phase sorting could be achieved by the use of retrospective gating either by an electrocardiogram, a pulse oximeter or by a self-gated MR sequence [47,48]. In addition, automated segmentations and thresholds could be implemented to guarantee reproducible measurements of perfusion maps [49]. Last but not least, whole brain perfusion measurements should be performed within a reasonable timeframe of 5 minutes like in pCASL MRI.

Despite the mentioned limitations of the current FREE brain perfusion method, it inherits the major advantage of being a very safe technique by avoiding the use of ionizing radiation or intravenous contrast media. This excludes rare, but possible complications like cancerogenic mutations by ionizing radiation or nephrogenic systemic fibrosis (NSF) and allergic reactions caused by the application of gadolinium-based contrast media [50–52]. In contrast to pCASL MRI, less distortion occurs due to susceptibility artifacts and no post labelling delay is needed, which could cause perfusion artifacts. However, broader clinical validation in terms of accuracy and reproducibility in larger patient cohorts needs to be investigated in future studies.

This study confirmed the similarity between the new technique and gadolinium enhancement MRI and CT perfusion only in the condition of brain metastases and large vessel occlusion. Its sensitivity for other diseases such as high-grade glioma, encephalitis, vasculitis or minor ischemic stroke is still uncertain. Also, this study did not consider the artifact effects of some patients (e.g., with V-P shunt).

See Table 2 for comparison of FREE, pCASL and contrast-based perfusion techniques.

The data suggest to possibly use a bSSFP sequence with a slice thickness of about 5mm for a reasonable balance between perfusion contrast and SNR and a flip angle of about 50–60 degrees. At 3T the SAR limits also need to be considered.

## Conclusions

In conclusion, we presented the feasibility to image brain perfusion in a healthy cohort and patients with brain metastases and acute stroke by exploiting flow-related enhancement in a bSSFP sequence. These results open up a new research field for this MRI technique for safe, fast and sensitive perfusion imaging in various pathologies within the brain.

## Supporting information

**S1 Fig. Frequency analysis of a FLASH sequence.** A 40˚ flip angle indicating a perfusion peak in the vessel (a) at the heart frequency close to 1 Hz (red triangle), but no prominent perfusion peak in the parenchyma ROI (b).
(TIF)

**S2 Fig. Perfusion-weighted FREE maps using a FLASH sequence at 3T with a flip angle of 40˚.** A temporal perfusion-weighted MIP is showing the maximal amplitude of the entire heart cycle for each voxel (a) and the perfusion-weighted map (b) is presenting the maximum median amplitude within the parenchyma ROI. Although showing strong perfusion contrast

of 77.80 x $10^{-3}$ in the blue vessel ROI only weak corticomedullary differentiation is visible. Perfusion contrast of 5.38 x $10^{-3}$ in the green parenchyma ROI.
(TIF)

**S3 Fig. Perfusion-weighted FREE map using a bSSFP sequence at 3T with a flip angle of 70˚.** A temporal perfusion-weighted MIP is showing the maximal amplitude of the entire heart cycle for each voxel (a) and the perfusion-weighted map (b) is presenting the maximum median amplitude within the parenchyma ROI. The perfusion contrast is 27.89 x $10^{-3}$ in the blue vessel ROI and 2.96 x $10^{-3}$ in the green parenchyma ROI.
(TIF)

**S4 Fig. Perfusion-weighted FREE map of a 78-year-old patient with an acute stroke.** A bSSFP sequence at 1.5T with a flip angle of 55˚. A temporal perfusion-weighted MIP is showing the maximal amplitude of the entire heart cycle for each voxel (left) and the perfusion-weighted map (right) is presenting the maximum median amplitude within the parenchyma ROI. Notice only minimal signal decrease in the temporal MIP, but clear demarcation of the stroke area in the parenchyma phase of the healthy hemisphere.
(TIF)

**S5 Fig. Correlation plots and Histograms of pCASL and FREE and FREE MIP.** Exemplary correlation plots of a healthy participant. Whereas the FREE histogram shows many zero values, the FREE MIP presents only few. This might be caused by the differing arrival time of the pule wave between grey and white matter and no perfusion signal in the white matter yet during the phase of the perfusion-weighted FREE map.
(TIF)

**S1 Table. MRI sequence parameters of the patients.**
(DOCX)

**S2 Table. MRI sequence parameters of the healthy.**
(DOCX)

**S3 Table. pCASL perfusion parameters of the healthy.**
(DOCX)

**S4 Table. CT perfusion parameters of the stroke patient.**
(DOCX)

**S1 Movie. Phase-resolved perfusion weighted movie of a patient with an acute stroke.** A bSSFP sequence at 1.5T with a flip angle of 55˚. Notice the phase delay within the vascular territory of the occluded left middle cerebral artery.
(ZIP)

**S1 Data.**
(XLSX)

## Acknowledgments

The authors would like to express their gratitude to the medical technical assistants from the Department of Radiology for their support with the MR measurements and patient care.

## Author Contributions

**Conceptualization:** Julian Glandorf.

**Data curation:** Julian Glandorf.

**Formal analysis:** Julian Glandorf, Filip Klimeš, Marcel Gutberlet.

**Methodology:** Agilo Luitger Kern, Jens Vogel-Claussen.

**Project administration:** Frank Wacker, Jens Vogel-Claussen.

**Software:** Julian Glandorf, Filip Klimeš, Andreas Voskrebenzev, Marcel Gutberlet, Agilo Luitger Kern.

**Supervision:** Frank Wacker, Jens Vogel-Claussen.

**Validation:** Nima Mahmoudi, Mike P. Wattjes.

**Visualization:** Julian Glandorf.

**Writing – original draft:** Julian Glandorf.

**Writing – review & editing:** Filip Klimeš, Andreas Voskrebenzev, Marcel Gutberlet, Agilo Luitger Kern, Norman Kornemann, Nima Mahmoudi, Mike P. Wattjes, Frank Wacker, Jens Vogel-Claussen.

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
