## [Decision Letter · Decision Letter 0]

7 Sep 2022

PONE-D-22-14162Feasibility of flow-related enhancement brain perfusion MRIPLOS ONE

Dear Dr. Julian Glandorf,

Thank you for submitting your manuscript to PLOS ONE. After careful consideration, we feel that it has merit but does not fully meet PLOS ONE’s publication criteria as it currently stands. Therefore, we invite you to submit a revised version of the manuscript that addresses the points raised during the review process.

We look forward to receiving your revised manuscript.

Kind regards,

Ayataka Fujimoto

Academic Editor

PLOS ONE

Journal Requirements:

Additional Editor Comments:

Please carefully see the comments from the reviewers and revise it in accordance with the comments.

Reviewers' comments:

Reviewer's Responses to Questions

**Comments to the Author**

1. Is the manuscript technically sound, and do the data support the conclusions?

Reviewer #1: Yes

Reviewer #2: Yes

2. Has the statistical analysis been performed appropriately and rigorously? 

Reviewer #1: Yes

Reviewer #2: Yes

3. Have the authors made all data underlying the findings in their manuscript fully available?

Reviewer #1: Yes

Reviewer #2: Yes

4. Is the manuscript presented in an intelligible fashion and written in standard English?

Reviewer #1: Yes

Reviewer #2: Yes

5. Review Comments to the Author

Reviewer #1: The authors present a novel method for measuring brain perfusion using MRI. This so-called FREE sequence does not require the administration of contrast medium and was originally developed to determine the perfusion of the lung parenchyma. The results are promising and can be used to measure brain perfusion. The work is highly relevant for clinical application. The manuscript is clearly written without substantial errors. I suggest publication of the manuscript after a minor revision.

1. The presented perfusion maps are similar to the classic CBF maps. Is it possible to give quantitative values of the blood flow. Does the methode allow to calculate the cerebral blood volume (CBV) based an the kinetics of the signal?

2. Two patients with metastasis of a melanoma were examined. Due to the content of melanin, these metastasis appear hyperintense on T1-weighted images without contrast media. Does this effect influence the perfusion contrast?

3. Fig. 4 b: Were the measurements performed before or after radiotherapy?

4. In the discussion section (in line 244) the dependency of the perfusion signal on the angle of blood flow is mentioned. In perspective, is it possible to quantify the directional dependence of perfusion using the method presented (e.g. in the sense of a perfusion tensor)?

5. In the stroke patient only the basal ganglia are hypointense on CT while the M1-segment is occludet. In such a case, the vessel is usually immediately recanalized by thrombectomy without delay by MRI. This should be commented on. What was the outcome of the patient. Did you perform a thrombectomy?

Reviewer #2: Summary of article:

Dr. Glandorf et al. have investigated the methodological validity of the new technique for flow-related enhancement brain perfusion MRI. The authors applied a method to describe perfusion-weighted maps without contrast media, which was reported in the research on lung MRI, into brain MRI. Specifically, the authors conducted three following experiments: (1) Determination of the best flip angles and slice thickness using the data of one health volunteer; (2) Correlation of classification of with-matter/gray matter between arterial spin labeling (ASL) method and their new method using the data of 25 healthy volunteer; (3) Investigation for the usefulness in pathologies using the data of three patients with cerebral metastases and one ischemic stroke patient. As a result, the authors found that (1) Perfusion contrast and its corticomedullary differentiation increased with flip angle; (2) classification in ASL and their new method was highly correlated (correlation coefficient: 0.36 in Pearson correlation test); (3) the results of the new technique were similar with the results of T1 weighted image with gadolinium enhancement. They concluded that their new technique without using contrast media would be safe and feasible.

Comments (Invitation on Aug 31, 2022, and comment submission on Sep 6, 2022)

This study addressed an interesting methodological paper for a new technique of enhancement brain MRI without using contrast media. This study could have a great impact on investigating less-invasive examinations. The authors’ motivation is reasonable because gadolinium enhancement MRI is known to cause rare but critical complications. I would like to congratulate all authors’ efforts in this paper. Please consider addressing some concerns as shown below.

Here are my comments and suggestions about this manuscript.

Major points:

[1] “Introduction”

This study consists of three experiments. Therefore, it is better to describe the purpose/aim and motivation of each experiment in the introduction, such as the followings:

[Aim 1]

[Aim 2]

[Aim 3]

[2] “Methods”

It is helpful for broad readers to add a brief explanation of what Sorensen-Dice coefficients means or how it is calculated in 1-2 sentence.

[3] “Results”

Please consider describing the time required for imaging in every experiment, although the authors show the required time only in a patient with an ischemic stroke.

[4] “Discussion”

It is better to summarize the characteristics or strengths/weaknesses of the new technique compared with those of ASL and gadolinium enhancement MRI in a table.

[5] “Discussion”

Please add the following limitations: (1) This study confirmed the similarity of the results between the new technique and gadolinium enhancement MRI only in the condition of brain metastases and large vessel occlusion. The feasibility for other diseases such as high-grade glioma, encephalitis, vasculitis, or minor ischemic stroke is still uncertain; (2) This study did not consider the artifact effects. Some patients (i.g. with V-P shunt) would have artifacts that would affect the MR imaging.

Minor points:

[6] “Figure 2”

Please add the explanation of (c) in the figure legend.

[7] “Results”

Please rewrite the subsection title “FREE vs pCASL” because this part does not show which is better or superior but the correlation/concordance of two different methods.

[8] “Results”

It is helpful to show the scatter plot of Pearson correlation.

[9] “Results”

Please explain or define what “Dice match” means. I guess that means the “spatial overlap assessed by Sorensen-Dice coefficients,” but it should be defined in the manuscript.

[10] “Results”

The explanation of (a)-(g) in Figure 5 should be in the figure legend, not in the main text.

6. PLOS authors have the option to publish the peer review history of their article (what does this mean?). If published, this will include your full peer review and any attached files.

Reviewer #1: **Yes: **Christian Ziener

Reviewer #2: **Yes: **Naoto Kuroda

---

## [Author Response · Author response to Decision Letter 0]

29 Sep 2022

Response to Reviewers

PONE-D-22-14162

Feasibility of flow-related enhancement brain perfusion MRI

PLOS ONE

Dear Dr. Fujimoto

the authors would like to thank you for the precise review of the manuscript and are grateful to submit a revised version based on the reviewers' comments. We feel, that these changes improve the quality of the manuscript. A point-by-point response to each comment can be found below.

Sincerely, 

Julian Glandorf

PLOS ONE

Journal Requirements:

Answer: The style requirements were applied.

Answer: An anonymized data set was uploaded as a Supporting Information file.

Comments to the Author

Reviewer #1: 

1. The presented perfusion maps are similar to the classic CBF maps. Is it possible to give quantitative values of the blood flow. Does the method allow to calculate the cerebral blood volume (CBV) based an the kinetics of the signal?

Answer: A quantification method has been developed in functional lung MRI, but has not been modified for brain perfusion yet, which is planned in a future study. However, this is quite challenging and will most likely produce a compound effect of cerebral blood volume, perfusion and CSF flow effects. The separation and proportion of these effects will have to be evaluated extensively in comparison to the ASL-method.

2. Two patients with metastasis of a melanoma were examined. Due to the content of melanin, these metastasis appear hyperintense on T1-weighted images without contrast media. Does this effect influence the perfusion contrast?

Answer: The perfusion contrast is most likely not influenced my melanin, as the perfusion contrast is generated by a subtraction so that the melanin-effect is cancelled out.

3. Fig. 4 b: Were the measurements performed before or after radiotherapy?

Answer: The measurements were performed after radiotherapy presenting lower perfusion contrast at that area: “Notice the differing perfusion contrast after radiation of the right frontal lobe in (b).”

4. In the discussion section (in line 244) the dependency of the perfusion signal on the angle of blood flow is mentioned. In perspective, is it possible to quantify the directional dependence of perfusion using the method presented (e.g. in the sense of a perfusion tensor)? 

Answer: Yes, I think so. You would need to image the specimen in three imaging planes and calculate the perfusion trajectory in each voxel depending on the height of the perfusion contrast in each plane. However, this would be time consuming.

5. In the stroke patient only the basal ganglia are hypointense on CT while the M1-segment is occludet. In such a case, the vessel is usually immediately recanalized by thrombectomy without delay by MRI. This should be commented on. What was the outcome of the patient. Did you perform a thrombectomy?

Answer: In this case there were 2 emergency patients at a time. This patient had an unknown time window and was examined by MRI to gain more information, while the other patient had a thrombectomy. Afterwards this patient had a thrombectomy as well, but showed demarked infarctions as already visible in the CT and MRI prior to thrombectomy. An additional comment was placed in the figure caption. 

Reviewer #2: 

Major points:

[1] “Introduction”

This study consists of three experiments. Therefore, it is better to describe the purpose/aim and motivation of each experiment in the introduction, such as the followings:

[Aim 1]

[Aim 2]

[Aim 3]

Answer: The recommendation was applied in the introduction. The three aims of the study were expressed. 

[2] “Methods”

It is helpful for broad readers to add a brief explanation of what Sorensen-Dice coefficients means or how it is calculated in 1-2 sentence.

Answer: An additional sentence with explanation was added. 

[3] “Results”

Please consider describing the time required for imaging in every experiment, although the authors show the required time only in a patient with an ischemic stroke.

Answer: The imaging took 230 seconds for one slice in each measurement in the healthy cohort and 127 seconds in the patient cohort. The information was added. 

[4] “Discussion”

It is better to summarize the characteristics or strengths/weaknesses of the new technique compared with those of ASL and gadolinium enhancement MRI in a table.

Answer: An additional Table (Table 2) was created for comparison of the different MRI brain perfusion techniques. 

[5] “Discussion”

Please add the following limitations: (1) This study confirmed the similarity of the results between the new technique and gadolinium enhancement MRI only in the condition of brain metastases and large vessel occlusion. The feasibility for other diseases such as high-grade glioma, encephalitis, vasculitis, or minor ischemic stroke is still uncertain; (2) This study did not consider the artifact effects. Some patients (i.g. with V-P shunt) would have artifacts that would affect the MR imaging.

Answer: The limitations were addressed in the discussion section. 

Minor points:

[6] “Figure 2”

Please add the explanation of (c) in the figure legend.

Answer: The reference was included. 

[7] “Results”

Please rewrite the subsection title “FREE vs pCASL” because this part does not show which is better or superior but the correlation/concordance of two different methods.

Answer: It was changed to “Correlation and spatial overlap of FREE and pCASL”.

[8] “Results”

It is helpful to show the scatter plot of Pearson correlation.

Answer: A plot was added as a Supplemental Figure (S6).

[9] “Results”

Please explain or define what “Dice match” means. I guess that means the “spatial overlap assessed by Sorensen-Dice coefficients,” but it should be defined in the manuscript.

Answer: An explanation is in the section “Correlation and spatial overlap between FREE and pCASL maps”

[10] “Results”

The explanation of (a)-(g) in Figure 5 should be in the figure legend, not in the main text.

Answer: The explanation was transferred into the figure legend. 

All figures were uploaded to PACE and modified accordingly.

---

## [Decision Letter · Decision Letter 1]

17 Oct 2022

Feasibility of flow-related enhancement brain perfusion MRI

PONE-D-22-14162R1

Dear Dr. Julian Glandorf,

We’re pleased to inform you that your manuscript has been judged scientifically suitable for publication and will be formally accepted for publication once it meets all outstanding technical requirements.

Kind regards,

Ayataka Fujimoto

Academic Editor

PLOS ONE

Additional Editor Comments (optional):

Dear authors,

Thank you very much for the revision.

Al reviewers accepted the latest version.

Reviewers' comments:

Reviewer's Responses to Questions

**Comments to the Author**

1. If the authors have adequately addressed your comments raised in a previous round of review and you feel that this manuscript is now acceptable for publication, you may indicate that here to bypass the “Comments to the Author” section, enter your conflict of interest statement in the “Confidential to Editor” section, and submit your "Accept" recommendation.

Reviewer #1: All comments have been addressed

Reviewer #2: All comments have been addressed

2. Is the manuscript technically sound, and do the data support the conclusions?

Reviewer #1: Yes

Reviewer #2: Yes

3. Has the statistical analysis been performed appropriately and rigorously? 

Reviewer #1: Yes

Reviewer #2: Yes

4. Have the authors made all data underlying the findings in their manuscript fully available?

Reviewer #1: Yes

Reviewer #2: Yes

5. Is the manuscript presented in an intelligible fashion and written in standard English?

Reviewer #1: Yes

Reviewer #2: Yes

6. Review Comments to the Author

Reviewer #1: All points have been carefully edited. The manuscript is clearly written. I recommend publication of the manuscript.

Reviewer #2: I appreciate the authors taking my suggestions and incorporating them into the manuscript.

I would like to congratulate their effort in the revision.

7. PLOS authors have the option to publish the peer review history of their article (what does this mean?). If published, this will include your full peer review and any attached files.

Reviewer #1: **Yes: **Christian Ziener

Reviewer #2: **Yes: **Naoto Kuroda

---

## [Editor Report · Acceptance letter]

28 Oct 2022

PONE-D-22-14162R1 

Feasibility of flow-related enhancement brain perfusion MRI 

Dear Dr. Glandorf:

I'm pleased to inform you that your manuscript has been deemed suitable for publication in PLOS ONE. Congratulations! Your manuscript is now with our production department. 

Kind regards, 

on behalf of

Dr. Ayataka Fujimoto 

Academic Editor

PLOS ONE